# BNT162b2 COVID-19 Vaccine Safety among Healthcare Workers of a Tertiary Hospital in Italy

**DOI:** 10.3390/vaccines11020477

**Published:** 2023-02-17

**Authors:** Flavia Beccia, Luca Regazzi, Eleonora Marziali, Viria Beccia, Domenico Pascucci, Nadia Mores, Giuseppe Vetrugno, Patrizia Laurenti

**Affiliations:** 1Section of Hygiene, Department of Life Sciences and Public Health, Università Cattolica del Sacro Cuore, 00168 Rome, Italy; 2Medical Oncology, Fondazione Policlinico Universitario Agostino Gemelli IRCCS, Università Cattolica del Sacro Cuore, 00168 Rome, Italy; 3Fondazione Policlinico Universitario Agostino Gemelli IRCCS, 00168 Rome, Italy; 4Department of Pharmacology, Faculty of Medicine, Università Cattolica del Sacro Cuore, 00168 Rome, Italy; 5Risk Management Unit, Fondazione Policlinico Universitario Agostino Gemelli IRCCS, 00168 Rome, Italy

**Keywords:** COVID-19, Comirnaty, BNT162b2, pharmacovigilance, pandemic, safety, vaccine, vaccination, public health, human

## Abstract

Millions of people have died because of the COVID-19 pandemic. The vaccination campaign helped tackle the pandemic and saved millions of lives. In a retrospective pharmacovigilance study, we explored the safety of the BNT162b2 (Comirnaty) vaccine among healthcare workers (HCWs) in a large Italian teaching hospital, and 2428 Adverse Events Reports (AERs) filed by HCWs after the administration of the first dose of vaccine were collected and analyzed, reporting the results quantitively and comparing them to the vaccine Summary of Product Characteristics (SPC). Spearman’s correlation coefficients were computed to investigate the correlation among reported adverse effects, and recurrent clusters of symptoms were investigated through the Principal Component Analysis (PCA) and k-means Cluster Analysis. The BNT162b2 vaccine’s safety profile was favorable, with predominant reports of early onset, mild, non-serious and short-term resolved symptoms. We observed higher than the expected frequency for various non-serious undesirable effects, especially among those listed and classified as less common in the SPC. Furthermore, we identified three clusters of adverse effects that were frequently reported together, defined by the presence/absence of fatigue, malaise, localized pain, chills, pyrexia, insomnia, nausea and injection site pain. Post-marketing pharmacovigilance activities, together with targeted public health interventions, can be valuable tools to promote vaccination and improve the control of the spread of the pandemic, especially in sensitive settings and populations such as hospitals and healthcare professionals.

## 1. Introduction

The pandemic caused by SARS-CoV-2 is an unprecedented public health emergency, resulting in high morbidity and mortality globally [1,2]. According to the World Health Organization (WHO), as of 22 January 2023, more than 664 million cases of SARS-CoV-2 have been diagnosed globally, resulting in more than 6.7 million deaths [3]. In Italy, as of 25 January 2023, there have been more than 25 million diagnosed cases of SARS-CoV-2 infection and nearly 187 deaths [4]. The Italian study on excess mortality found more than 108,000 excess deaths (from all causes) in the March–December 2020 period compared with the 2015–2019 five-year average [5,6].

To counter the spread of the pandemic, public health measures such as physical distancing, quarantine and isolation were implemented [2,7,8]. In addition, great efforts have been made to develop and administer the vaccine [9,10], which has proven to be an effective tool for preventing severe forms of the disease [8,11].

After an evaluation of the safety and efficacy of the BNT162b2 mRNA vaccine (Pfizer-BioNTech, Comirnaty) [12,13,14], in December 2020, the Food and Drug Administration (FDA) [15] and the European Medicines Agency (EMA) [16] authorized its emergency use and administration to persons over the age of 16 years, following a schedule of two doses 21 days apart for the primary course. After the approval by the Italian Medicines Agency (AIFA) [17], the vaccination campaign was started in Italy on 27 December 2020 [18]. The administration followed a priority order based on the risk of infection and severe disease for COVID-19 in different population groups, and HCWs were the first target audience [6,19,20,21], both to ensure protection and safety for them and patients and for the sustainability of healthcare systems [12,13,14]. Vaccination was then extended to other population groups and to all age groups [6]. As of 31 January 2022, a total of 128 million doses of the COVID-19 vaccine have been administered in Italy, 64% of them with the Comirnaty vaccine [6]. It is estimated that, as a result of vaccination, more than 470,000 deaths in Europe and more than 290,000 hospitalizations and about 77,000 deaths in Italy were prevented in 2021 [6]. These data show the effectiveness of vaccines in preventing severe or critical forms of COVID-19 and in leading to a reduction in the impact of the pandemic on care services [6,22,23,24].

Although rapid and mass vaccination is essential in combating severe forms of COVID-19, hesitation toward the vaccine has been observed, including by HCWs, due to the fear of adverse reactions after immunization [25,26,27]. Therefore, continuous evaluation and monitoring of adverse events are critical to demonstrate the safety of vaccines [28,29,30] and can be crucial to the success of an immunization program [31]. The WHO defines adverse events following immunization (AEFI) as “Any untoward medical event that follows immunization and that does not necessarily have a causal relationship with the usage of the vaccine” [30,32,33]. AEFIs can be: “Vaccine product-related”; “Vaccine quality-related”; “Immunization error-related”; “Immunization stress-related”; “Coincidental” [32,33,34]. In addition, an adverse event of special interest (AESI) is defined as “A preidentified and predefined medically-significant event that has the potential to be causally associated with a vaccine product that needs to be carefully monitored and confirmed by further specific studies” [32]. AESIs include serious but rarely fatal or permanent events, such as allergic reactions or seizures [35]. The surveillance of AEFIs and AESIs can be conducted by applying a passive approach, based on spontaneous and unsolicited reports [36,37], or an active approach, which involves intentionally searching for AEFIs [38] and can result in a more precise identification of the event [14,39]. In Italy, adverse reactions are voluntarily reported by vaccine recipients to the COVID-19 vaccine pharmacovigilance surveillance established by the AIFA [40], which publishes monthly reports on each vaccine [8]. Clinical trials on mRNA vaccines have reported transient vaccine reactogenicity, with mostly mild-to-moderate adverse events, more frequently after the second dose than the first dose and among participants younger than 65 years of age compared with older participants [41,42,43,44]. However, rare serious adverse events have also been reported in clinical trials among BNT162b2 recipients [42,45,46,47,48,49].

To support and improve immunization strategies, it is necessary to collect information on vaccine safety in real-world scenarios, as well as have a follow-up system of adverse events once they are reported Therefore, we conducted a retrospective observational pharmacovigilance study to evaluate the safety of the first dose of the BNT162b2 vaccine during the first phase of the vaccination campaign against SARS-CoV-2 (between 28 December 2020 and 15 March 2021) at Fondazione Policlinico Universitario “A. Gemelli” IRCCS (FPG), a large teaching hospital in Rome, Italy.

## 2. Materials and Methods

The sample was composed of all the HCWs vaccinated with the first dose of BNT162b2 vaccine between 28 December 2020 and 15 March 2021. The data were searched in September 2021 and extracted from the national pharmacovigilance network—RNF, adopting as filters sender, the single healthcare structure, vaccination date, vaccine and source, in order to select only spontaneous and/or solicited reports from HCWs. Cases reported by hospital doctors were not extracted or included in the analysis. For the analysis, we considered the following: number of cases, demographic characteristics of the patients, seriousness and symptoms reported. No causality assessment has been performed and/or evaluated. The terms for adverse reactions were coded as Preferred Terms (PT), according to the Medical Dictionary for Regulatory Activities (MedDRa), versions 24.0 (1) and 25.0 (1). The reported symptoms were compared for frequency to the information available at point 4.8 of the Summary of Product Characteristics (SPC) for Comirnaty vaccine (at the analysis date, November 2022, the last available was “Comirnaty Original—OmicronBA.1—06/10/2022”), where the adverse reactions are categorized by System Organ Classification (SOC) and Preferred Term (PT, i.e., a distinct descriptor or single medical concept for a symptom, sign and disease diagnosis) according to the frequency of occurrence as the following: very common (i.e., >10%), common (i.e., between 1% and 10%), not common (i.e., between 0.1% and 1%), rare (i.e., <0.1%), very rare (i.e., <0.01%) and not known [50].

Categorical data are presented as absolute and relative numbers of patients. For continuous data, mean and standard deviation (SD) were used, depending on data distribution.

Spearman’s correlation coefficients (r_s_) were computed to investigate the correlations among reported adverse events.

Given the high number of variables (i.e., reported adverse events) in the dataset, a Principal Component Analysis (PCA) was realized to reduce the dimensionality and investigate groups of associated adverse events [51]. PCA is a statistical method to identify the ways in which numeric variables covary, which produces as outcome a smaller set of variables that retain most of the variability of the full dataset. These are called principal components (PCs) and are composites of the original variables multiplied by a set of weights.

Subsequently, to assess the predictive utility of the identified PCs, the individual adverse events reports were clustered in the principal components space using k-means clustering, a clustering technique that divides the data into k clusters by minimizing the sum of the squared distances of each record to its assigned cluster [52]. The k number of clusters was set equal to 3 after iteratively testing the method with multiple k values, as k = 3 clusters are able to provide the best balance between higher number of clusters with substantially different underlying PCs profiles and better discrimination between clusters.

All statistical analyses were conducted with a statistical significance level set at *p* < 0.05 and performed with STATA 14 (StataCorp LP, College Station, TX, USA) and R 4.0.3 (The R Foundation for Statistical Computing, Vienna, Austria).

The study protocol was approved by the FPG ethical committee (ID3973).

## 3. Results

Two-thousand four hundred and twenty-eight adverse events reports (AERs) were collected from five thousand one hundred and sixty-two vaccinated HCWs, one thousand six hundred and ninety-one (69.65%) who reported were female. The mean age was 34.05 (SD 12.31, range 18–65).

Additionally, 29.28% of subjects reported only one symptom, whereas 70.72% reported more than one. Eighty-five percent of the AERs were reported from HCWs under 55 years of age.

All the patients reported non-serious adverse reactions in the same day of the vaccination, with a complete resolution within seven days, and 59.93% of AERs were collected in January, 32.54% in February and 6.51% in March. Only 25 cases occurred between the 28th and 31st of December 2020.

Thirty different symptoms were reported. The distribution of all the reported symptoms is reported in Table 1, comparing the data to the SPC Comirnaty at 4.8.

Among the most common reactions, fever was reported by 17% of respondents, whereas, in the SPC, it appeared to be less frequent, nearly 10%. Fever was often accompanied by chills (21%) and fatigue (48%). Headache occurred in 28% of respondents, whereas it was expected to occur in more than 50% of cases. Diarrhea occurred in 0.21% of subjects. This symptom was not observed during the authorization trials. The exact percentage of the reference is not known, although it is enlisted among the very common adverse reactions. Arthralgia and myalgia were referred to by 0.12% and 0.04% of individuals, respectively, far less than expected. Tachycardia was found in 0.12% of cases (ref >60%). Injection site pain was found in 64% of cases and was the most reported adverse reaction. In addition, injection site swelling was found to be rare (0.04%). Localized pain was common (37%) and not listed as a separate term in the SPC for SOC General disorders and administration site conditions.

As for the common reactions, our findings highlighted higher rates of occurrence for nausea, which was reported by 14% of respondents. Vomiting and injection site redness were found in 0.04% of cases.

Some symptoms listed as “uncommon” in the SPC were reported in the study population to a much higher extent than expected: hypersensitivity reactions (e.g., rash and pruritus) were present in 6.7% of reported symptoms, lymphadenopathy in 9% of cases, insomnia in 20% and malaise in 38%. On the other hand, as expected from the SCP, asthenia was reported in 0.12% of cases, whereas hyperhidrosis and night sweats accounted for 0.08% and 0.12% of cases, respectively. Lethargy, decreased appetite, pain in extremities and injection site pruritus were not observed. Unlisted and not better specified gastrointestinal disorders were reported in 0.12% of cases.

Among the SPC-listed rare reactions, acute peripheral facial paralysis was not observed. On the other hand, unexpected events have been reported: sporadic cases of photophobia (*n* = 2, 0.08%), sinus congestion (*n* = 2, 0.08%), cough (*n* = 2, 0.08%), presyncope (*n* = 1, 0.04%), aphtha (*n* = 1, 0.04%) and pustule (*n* = 1, 0.04%) were reported. There were no cases of myocarditis, pericarditis, facial swelling, extensive swelling of the vaccinated limb, erythema multiforme and anaphylaxis. Six cases of paranesthesia (0.25%) and one (0.04%) of hypoesthesia occurred. In particular, one case of paranesthesia was reported in one arm, while the remaining cases and hypoesthesia were reported in the facial area.

Through Spearman’s correlation analysis, a cluster of interrelated reported adverse events was identified, including pyrexia, chills, malaise, fatigue, headache, insomnia, nausea and localized pain in sites other than the injection site. Within this cluster, the stronger correlation coefficients were found for the following pairs of adverse events: fatigue ~ malaise (rs = 0.45), malaise ~ localized pain (rs = 0.37), pyrexia ~ chills (rs = 0.36), chills ~ malaise (rs = 0.34) and fatigue ~ localized pain (rs = 0.33). All the members of this cluster also showed a positive, though very weak, association with lymphadenopathy. Conversely, a weak negative association was described for injection site pain with malaise (rs = −0.22), pyrexia (rs = −0.22), chills (rs = −0.22), nausea (rs = −0.17), insomnia (rs = −0.16) and lymphadenopathy (rs = −0.12).

Furthermore, other isolated correlations were identified, including gastrointestinal disorders ~ aphtha (rs = 0.58), gastrointestinal disorders ~ injection site swelling (rs = 0.58), sinus congestion ~ arthralgia (rs = 0.41), tachycardia ~ hypoesthesia (rs = 0.58), tachycardia ~ sweats (rs = 0.41), tachycardia ~ hyperidrosis (rs = 0.41), tachycardia ~ diarrhea (rs = 0.26), hyperhidrosis ~ diarrhea gastrointestinal disorders ~ aphtha (rs = 0.58) and gastrointestinal disorders ~ injection site swelling (rs = 0.32).

The complete results of the Spearman’s correlation analysis are included in Appendix A as a correlation matrix, with correlation coefficients (rs) included in the matrix cells and an additional color coding of the cells to graphically summarize the strength and direction of each correlation.

The Principal Component Analysis (PCAs) identified a total of 28 principal components (PCs) overall. However, the first three components were selected as the most relevant ones, as, together, they can explain up to 51.7% of the variance (PC1 = 28.7%, PC2 = 13.4% and PC3 = 9.6%; see Appendix A for the proportion of variance explained by PC4-28).

Figure 1 graphically represents the contribution of each different vector (i.e., each reported adverse effect) on PC1 and PC2 (see also Appendix A for an alternative representation). The project values of each vector on each PC show how much weight they have on that PC, while the angles between the vectors explains how characteristics correlate with one another (0° = maximum positive correlation, 90° = no correlation and 180° = maximum negative correlation). PC1 was positively associated with injection site pain, while an opposite relation was found for malaise, fatigue, localized pain, headache, pyrexia, insomnia, nausea and lymphadenopathy (in descending order of importance). PC2 was strongly and negatively associated with injection site pain but also received a negative contribution from fatigue, localized pain and malaise (in descending order of importance). On the other hand, PC2 resulted positively associated with chills, insomnia, pyrexia, nausea and lymphadenopathy (in descending order of importance).

The analysis of the angles between vectors provides some further information. Injection site pain vector evidently behaves differently from all the other adverse events vectors, with a decreasing degree of dissimilarity from insomnia to fatigue (anticlockwise), confirming the negative correlations observed in the previously described analysis. On the other hand, fatigue, localized pain and malaise appeared to be more strongly interrelated with one another than with the other vectors, confirming results from the correlation analysis. Similarly, chills, pyrexia and nausea resulted strongly related with one another and, weaker, with insomnia, lymphadenopathy and headache, once again confirming the results from the previous Spearman’s correlation analysis.

As for PC3, analysis of the PCA eigenvectors (see Appendix A) showed that it was negatively associated with headache, insomnia, injection site pain, pyrexia, hypersensitivity, nausea, lymphadenopathy and localized pain (in descending order of importance) and positively associated with malaise, fatigue and chills (in descending order of importance).

The k-means cluster analysis of principal components allowed to identify three clusters of reports, graphically represented on the PC1–PC2 bidimensional space in Figure 2. The analysis of the clusters’ profiles in terms of PCs’ contributions (see Appendix A) showed that Cluster 1 was negatively associated with PC1 (strongly), PC2 (moderately) and PC3 (very weakly). The same analysis demonstrated that Cluster 2 was positively associated with PC2 (strongly) and PC3 (weakly), as well as negatively associated with PC1 (very weakly). Finally, Cluster 3 had a strong positive association with PC1 and a negative association with PC2 (moderate) and PC3 (weak).

Putting together the results from the k-means cluster analysis of the principal components with the results from the PCA of the reported adverse events, it was possible to describe a characterizing symptoms profile for each cluster of reports.

Cluster 1 (−PC1 > −PC2 > −PC3) was mainly characterized by the frequent presence of fatigue (−PC1 ~ −PC2, as, for this variable, they give contributions in the same direction), malaise (−PC1 ~ −PC2), localized pain (−PC1 ~ −PC2), chills (−PC1 >< −PC2, as, for this variable, they give contributions in opposite directions but the weight of PC1 is stronger), pyrexia (−PC1 >< −PC2), insomnia (−PC1 >< −PC2) and nausea (−PC1 >< −PC2). On the other hand, Cluster 1 was also characterized by a lower frequency of injection site pain (−PC1 >< −PC2).

Cluster 2 (+PC2 > +PC3 > −PC1) was mainly characterized by the frequent presence of chills (+PC2 ~ +PC3), insomnia (+PC2 >< +PC3), pyrexia (+PC2 >< +PC3) and nausea (+PC2 >< +PC3) and by a lower frequency of injection site pain (+PC2 ~ +PC3), localized pain (+PC2 ~ +PC3), fatigue (+PC2 >< +PC3) and headache (+PC3).

Finally, Cluster 3 2 (+PC1 > −PC2 > −PC3) was mainly characterized by the higher presence of injection site pain (+PC1 >< −PC2 ~ PC3) and by the lower presence of all other adverse events (+PC1 ~ −PC2 >< −PC3 or +PC1 ~ −PC3 >< −PC2), with the main exception of localized pain (+PC1 >< −PC2 ~ −PC3).

These results were confirmed by the descriptive analysis of reported adverse events frequency across the three clusters (Appendix A).

## 4. Discussion

This study analyzes AERs collected in relation to the administration of the first dose of the BNT162b2 COVID-19 vaccine to HCWs in an Italian tertiary care hospital. To our knowledge, this is the first attempt to systematically explore the occurrence of adverse events following the COVID-19 first dose of vaccination with BNT162b2 (Comirnaty) in a specific population of HCWs. The Comirnaty vaccine is the most widely used worldwide and the one for which proportionally more spontaneous reports of adverse events have been communicated after immunization. Based on the analysis of these reports, the SPC has been updated to reflect changes in the frequency and type of undesirable effects, reiterating the importance of post-marketing surveillance of medicinal products.

The most frequent symptoms were non-serious, with rapid onset and resolved within a few days, being mostly explained by the activation of the immune system in response to the vaccination and in line with the vaccine’s safety profile, as described in SPC.

The data collected showed a safety profile qualitatively similar to that expected on the basis of the results from the two main safety studies by Polack and colleagues and the periodic safety report produced by the Italian pharmacovigilance agency AIFA and the European Medicine Agency—EMA [42,53]. However, the reporting rate of AERs was significantly higher in our sample, as we recorded a rate of 47,000 AERs per 100,000 doses administered compared to the expected rate of 100. This aspect could be related to the vaccine team’s focus on raising awareness about the importance of reporting events and strongly soliciting and supporting reporting. It also provides further evidence of the extent of the underreporting of adverse events in post-marketing pharmacovigilance, if not proactively enforced. Regarding the frequency of adverse events described in the SPC and the literature data, our results showed a complex landscape of expected and novel symptoms. Differences in the expected and observed frequencies could be partly explained by the characteristics of the population, which also includes comorbid or immunocompromised individuals or those previously infected with COVID-19 [54].

Although the numerosity of AERs is substantial for the time frame of the reporting period, even compared with the experiences of other vaccinating centers, the number does not give a clear perspective on rare and very rare events, potentially leading to misleading frequencies [14,55,56,57,58]. In addition, AERs are often biased by the absence of conclusive evidence of a causal relationship between exposure to the medicinal product and the occurrence of the reported event, as well as by reporting and classification errors and coding issues [59,60].

Based on the identified symptoms and the significant proportion of the sample (70%) reporting more than one symptom, the association between the different symptoms was analyzed through Spearman’s correlation and further explored through the Principal Component Analysis and k-means cluster analysis. Three main clusters were identified, which were characterized, respectively, by (1) the frequent association of fatigue, malaise, localized pain, chills, pyrexia, insomnia and nausea; (2) the frequent association of chills, pyrexia, insomnia and nausea but not of fatigue, malaise and localized pain and (3) the isolated presence of injection site pain.

It should be kept in mind that the analysis of spontaneous reporting could be affected by coding limitations. Such codings can be inaccurate, as is the case for localized pain (not at the injection site), which could be a proxy for other general musculoskeletal symptoms (e.g., arthralgia or myalgia), which were instead underreported. Similarly, it should be noted that a relatively smaller weight was probably given to injection site pain, as this symptom was largely expected. In other words, injection site pain might have been underestimated in AERs and especially in those presented by people who reported more severe symptoms. This may explain the underreporting of this symptom, which is expected to be the most frequent, as well as its separate clustering with respect to all other symptoms, under the assumption that it might have been reported when it was the only symptom and not reported when more severe symptoms were present.

In our experience, lymphadenopathy was significantly over-reported with respect to the frequency expected from the SPC. In this respect, it must be noted that the opportunity of a more detailed coding of the symptom “lymphadenopathy” into “systemic lymphadenopathy” and “localized lymphadenopathy” might be useful. It would allow to better distinguish between a systemic issue (which is expected to be uncommon) or one limited to the injection site (which is expected to be much more frequent). As a matter of fact, the considerable local occurrence of localized lymphadenopathy might unnecessarily alarm a vaccinated person. For this reason, a separate piece of information about the occurrence of localized lymphadenopathy should be provided on the SPC, as this would support immunizers to best inform their patients during the collection of informed consent.

Various rare adverse events were reported, both expected (i.e., hypoesthesia, paranesthesia) and unexpected (i.e., sinus congestion, coughing, photophobia and presyncope). To better characterize such reactions it would be useful to assess the health conditions and medical history of the vaccinated subjects over time, i.e., pre- and post-vaccination, and the evolution of adverse symptomatology by means of a follow-up system. However, this kind of assessment is often not possible on a large scale, without manually accessing each single report. In fact, the lack of compulsory, structured fields for clinical information (beside PT coding) inevitably leads to a loss of detail at the moment of data retrieval. Thus, implementing a follow-up system as a part of the online reporting platform could provide a tool to identify those patients whose symptoms should be followed over time (for example, by asking the reporter himself for updates on the condition through the contact details released at the time of the report).

Furthermore, based on the information gathered from our study, it might be useful to dynamically identify profiles, or “phenotypes”, of disease from individual report to facilitate the subsequent reporting of adverse events. This might facilitate signal identification, allowing for effective identification and response to safety concerns, as well as informing the development of effective policies and strategies [61,62,63]. Together with a follow-up system, the tailored adaptation of the reporting system to individual reports submitted by patients might allow pharmacovigilance and prevention institutions to identify and respond to any health concerns that may arise more promptly. By tracking and monitoring patients’ reports, the system could be able to detect patterns or trends in patient concerns and complaints to inform the development of targeted interventions and risk management plans. Additionally, these interventions could help to increase patient engagement and trust in the reporting system, which can in turn lead to more accurate and exhaustive reporting. Overall, incorporating a robust follow-up system into the adverse event reporting system seems essential for addressing perceived health concerns and providing reporters and patients with the necessary support and resources to address their specific needs.

## 5. Conclusions

COVID-19 has shaped the world as we know it, and COVID-19 vaccination, given the increasing possibility of it becoming a seasonal vaccination, is still an ever-present issue to be researched [64]. We conducted a retrospective pharmacovigilance study on the safety of the BNT162b2 (Comirnaty) vaccine among healthcare workers (HCWs) in a large Italian teaching hospital. The BNT162b2 vaccine’s safety profile was overall favorable, with predominant reports of early onset, mild, non-serious and short-term resolved symptoms. However, the frequency of various non-serious undesirable effects was higher than expected, especially among those classified as less common in the SPC. The results of the study suggest that post-marketing pharmacovigilance activities can be valuable tools to promote vaccination and improve the control of the spread of the pandemic, especially in sensitive settings and populations such as hospitals and healthcare professionals. Overall, the results highlight the importance of vaccine safety assessment, also through the pharmacovigilance activities on real-world data, in helping public health officials and vaccine manufacturers to identify and investigate potential vaccine safety issues. In fact, based on the data collected, pharmacovigilance and public health institutions and health systems can adapt information strategies and reporting systems to improve vaccine safety awareness and encourage more people to get vaccinated.

## Figures and Tables

**Figure 1 vaccines-11-00477-f001:**
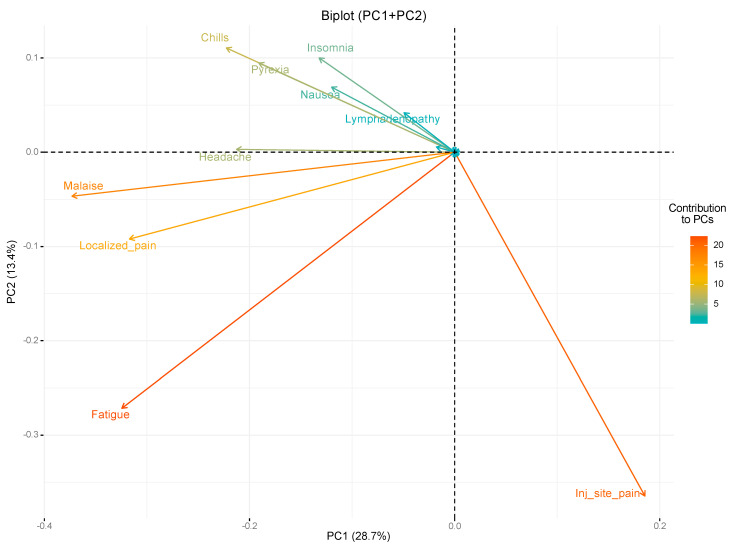
Biplot between PC1 and PC2 showing contribution of different vectors (i.e., reported adverse events) on the identified principal components. Only vectors that provide stronger contributions are reported.

**Figure 2 vaccines-11-00477-f002:**
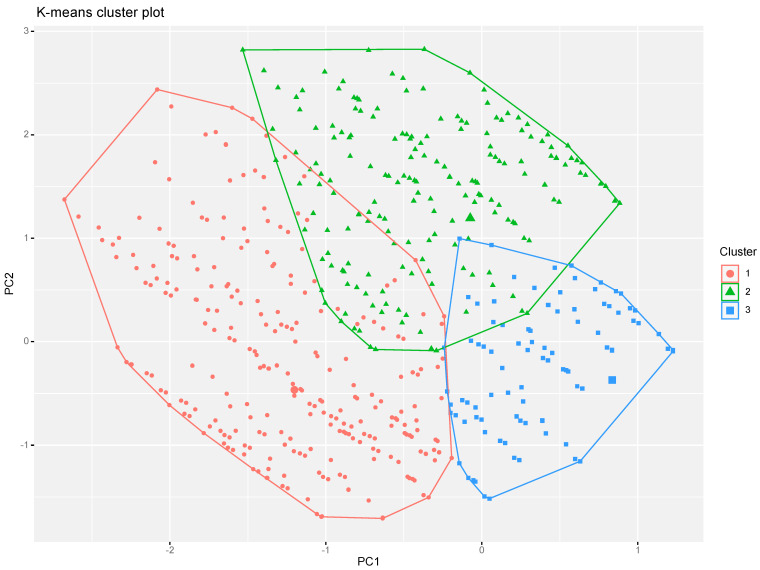
Graphical representations of reports clusters identified by means of k-means cluster analysis on the identified principal components.

**Table 1 vaccines-11-00477-t001:** Detailed FPG safety results compared to the SPC of Comirnaty (COMIRNATY_ORIGINAL_OMICRON_BA.1_06.10.2022).

System Organ Class	Very Common (≥1/10)	Common (≥1/100 to <1/10)	Uncommon (≥1/1000 to <1/100)	Rare (≥1/10,000 to <1/1000)	Very Rare (<1/10,000)	Not Known ^§^
Blood and lymphatic system disorders	–	–	Lymphadenopathy **9.10%**	–	–	–
Immune system disorders	–	–	Hypersensitivity reactions (e.g., rash, pruritus,) **6.71%**	Hypersensitivity reactions (urticaria, angioedema) NR	–	Anaphylaxis NR
Metabolism and nutrition disorders	–	–	Decreased appetite NR	–	–	–
Psychiatric disorders	–	–	Insomnia **20.06%**	–	–	–
Nervous system disorders	Headache 27.68% (ref. >50%)	–	Lethargy NR	Acute peripheral facial paralysis NR **Photophobia** **0.08%** **Presyncope** **0.04%**	–	Paresthesia **0.25%** Hypoaesthesia **0.04%**
Cardiac disorders	Tachycardia **0.12%** (ref. >60%)	–	–		Myocarditis; Pericarditis NR	
Respiratory disorders *	–	–	–	**Sinus congestion** **0.08%** **Cough** **0.08%**	–	–
Gastrointestinal disorders	Diarrhea **0.21%**	Nausea **13.96%** Vomiting **0.04%**	**Gastrointestinal disorders** **0.12%**	–	–	–
Skin and subcutaneous tissue disorder			Hyperhidrosis **0.08%** Night sweats 0.12%	**Aphtha** **0.04%**	–	Eritema multiforme NR
Musculoskeletal and connective tissue disorders	Arthralgia **0.12%** (ref. >20%) Myalgia **0.04%** (ref. >40%) **Localized pain** **36.90%**	–	Pain in extremity NR	–	–	–
General disorders and administration site conditions	Injection site pain 63.92% (ref. >80%) Fatigue 48.23% (ref. >60%) Chills 21.29% (ref. >30%) Pyrexia 16.89 (ref. >10%) Injection site swelling **0.04%** (ref.>10%)	Injection site redness **0.04%**	Malaise **37.77%** Injection site pruritus NR Asthenia 0.12%	**Pustule** **0.04%**	–	Extensive swelling of vaccinated limb NR Facial swelling NR

**§** indicates that frequencies cannot be estimated from the available data. **–** indicates that no symptoms were expected for those categories. ***** indicates a new line inserted on the basis of our results, as respiratory disorders were not reported in the SPC. The percentages that do not align with the category were highlighted in **bold**, as well the unexpected symptoms detected.

## Data Availability

The data that support the findings of this study are available on request from the corresponding author (L.R.) or the first author (F.B.).

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
