# Peer review of "BNT162b2 COVID-19 Vaccine Safety among Healthcare Workers of a Tertiary Hospital in Italy"

_vaccines, 2023, doi:10.3390/vaccines11020477_

Round 1

Reviewer 1 Report

This paper aims to describe the adverse events of Pfizer vaccine (BNT152b2) in the real world. The population involved is a non – negligible number of health care workers.  The topic is original and addresses a very interesting subject, the adverse effects of a vaccine for a lethal and dangerous disease. Evaluating a new vaccine in the real world is fundamental so that people hesitating to take the vaccine are not concerned with adverse events which do not exist. It is one of the first studies of the vaccines’ adverse events in a population of health care workers. The introduction is satisfactory and clearly describing the background of the study. Methods are well presented. No controls are needed in this type of article 

Discussion: Line 134 , what is FPG? 

The discussion in several points needs to be checked for too long sentences and easier to understand meanings. If you make shorter sentences , the meaning is going to be better understood. Examples lines 304 - 307 and 346- 350.  

More emphasis to the results presented should be given instead of general remarks. 

Author Response

We thank the reviewer for their useful suggestions. We detail our answers hereafter.

Discussion: Line 134 , what is FPG? 

The acronym FPG is defined at the end of the Introduction section (line 94).

The discussion in several points needs to be checked for too long sentences and easier to understand meanings. If you make shorter sentences, the meaning is going to be better understood. Examples lines 304 - 307 and 346- 350.  More emphasis to the results presented should be given instead of general remarks. 

We have edited the Discussion and the Conclusion sections as requested.

Reviewer 2 Report

Comments for the manuscript:

1.      Define the timing of vaccine administration at the end of introduction section

2.      Detail the type and timing of vaccination in Materials and Methods section

3.      Define FPG ethical committee

4.      Include the age of recruited people (minimum and maximum)

5.      Define the ICRS

6.      Consider to integrate the description of cohort underlining the possibility of previous infection of subjects before the vaccination, and the influence on symptoms

7.      Integrate the bibliography with studies of safety using other COVID vaccines, if available

8.      Check the punctuation in the whole manuscript

Author Response

We thank the reviewer for their useful suggestions. We detail our answers hereafter.

  1. Define the timing of vaccine administration at the end of introduction section

We have detailed the timing of the vaccination schedule at lines 50-51 and the evaluated period at lines 92-93.

  1. Detail the type and timing of vaccination in Materials and Methods section

We have detailed the type (BNT162b2), dose (first) and the evaluated period (28 December 2020 and 15 March 2021) at the beginning of the Methods section.

  1. Define FPG ethical committee

The acronym FPG was already defined at the end of the introduction (line 92 of the original manuscript).

  1. Include the age of recruited people (minimum and maximum) 

We have added the age range at lines 148-149.

  1. Define the ICRS 

We have replaced “ICRS” with the already defined acronym AERs (Adverse Events Reports).

  1. Consider to integrate the description of cohort underlining the possibility of previous infection of subjects before the vaccination, and the influence on symptoms

Unfortunately, the national pharmacovigilance network – RNF hosted by the Italian Medicines Agency (AIFA) only provides anonymized data without information concerning previous infection.

  1. Integrate the bibliography with studies of safety using other COVID vaccines, if available

We think the bibliography we have provided is sufficiently extensive and up to date. As this is not a comparative study, we do not think further bibliography concerning other COVID-19 vaccines is appropriate.

  1. Check the punctuation in the whole manuscript

We have checked the punctuation in the whole manuscript.

Reviewer 3 Report

I was invited to revisethe paper entitled "BNT162b2 COVID-19 Vaccine Safety among Healthcare Workers of a tertiary hospital in Italy". It aimed to evaluate the safety of BNT162b2 vaccine among hcw.

The topic is important for public health. The evaluation of safety and efficacy of a vaccine is foundamental, and pharmacovigilance data from real world are necessary to improve the knowledge on this field.

Introduction deeply describe the study background and methods section was clearly presented. In particular PCA was an innovative way to evaluate associated AEs.

Reasult appear clear and easy to read.

I have only some minor observations:

- Did Authors considered the history of priory Covid infection?

- Do Authors have data on hospital admission after vaccination? recent paper from Italy showed the lack of association between covid-19 vaccination and hospitalization for severe AEs. This point should be discussed;

- Authors should also consider the number of the dose in the analysis.

Author Response

We thank the reviewer for their useful suggestions. We detail our answers hereafter.

- Did Authors considered the history of priory Covid infection?            

Unfortunately, the national pharmacovigilance network – RNF hosted by the Italian Medicines Agency (AIFA) only provides anonymized data without information concerning previous infection, since this piece of information is not requested in the pharmacovigilance official form.

- Do Authors have data on hospital admission after vaccination? recent paper from Italy showed the lack of association between covid-19 vaccination and hospitalization for severe AEs. This point should be discussed;

According to the European Medicines Agency (EMA), a serious adverse event (experience) or reaction is any untoward medical occurrence that at any dose: result is n death, is life-threatening, requires inpatient hospitalisation or prolongation of existing hospitalization, results in persistent or significant disability/incapacity, or is a congenital anomaly/birth defect.

In our sample, so serious adverse events were reported. As such, among reported adverse events, none required hospitalization nor had any long-lasting negative health outcomes.

Moreover, as the national pharmacovigilance network – RNF hosted by the Italian Medicines Agency (AIFA) only provides anonymized data, we could not access separately any information related to hospitalizations that might have happened after the adverse events were reported.

- Authors should also consider the number of the dose in the analysis.

We have considered only first doses. We have made it clearer both in the Introduction (line 91) and in the Methods (line 96).